# An Artificial Intelligence Application for Post-Earthquake Damage Mapping in Palu, Central Sulawesi, Indonesia

**DOI:** 10.3390/s19030542

**Published:** 2019-01-28

**Authors:** Mutiara Syifa, Prima Riza Kadavi, Chang-Wook Lee

**Affiliations:** Division of Science Education, Kangwon National University, Chuncheon-si, Gangwon-do 24341, Korea; mutiarasyifa@kangwon.ac.kr (M.S.); rizakadavi@gmail.com (P.R.K.)

**Keywords:** ANN, Palu earthquake, post-earthquake damage map, SVM

## Abstract

A Mw 7.4 earthquake hit Donggala County, Central Sulawesi Province, Indonesia, on 28 September 2018, triggering a tsunami and liquefaction in Palu City and Donggala. Around 2101 fatalities ensued and 68,451 houses were damaged by the earthquake. In light of this devastating event, a post-earthquake map is required to establish the first step in the evacuation and mitigation plan. In this study, remote sensing imagery from the Landsat-8 and Sentinel-2 satellites was used. Pre- and post-earthquake satellite images were classified using artificial neural network (ANN) and support vector machine (SVM) classifiers and processed using a decorrelation method to generate the post-earthquake damage map. The affected areas were compared to the field data, the percentage conformity between the ANN and SVM results was analyzed, and four post-earthquake damage maps were generated. Based on the conformity analysis, the Landsat-8 imagery (85.83%) was superior to that of Sentinel-2 (63.88%). The resulting post-earthquake damage map can be used to assess the distribution of seismic damage following the Palu earthquake and may be used to mitigate damage in the event of future earthquakes.

## 1. Introduction

On 28 September 2018, at 17:02 UTC + 7, a Mw 7.4 earthquake hit Donggala County in Central Sulawesi, Indonesia, as reported by the National Board for Disaster Management (BNPB) of Indonesia [1]. The earthquake’s epicenter was located around 26 km northeast of Donggala (0.20 S, 119.89 E); the earthquake occurred at a depth of 11 km as a result of strike–slip faulting of the Sesar Palu-Koro fault at shallow depths within the interior of the Molucca Sea microplate, part of the broader Sunda tectonic plate [2]. Focal mechanism solutions for the earthquake indicated that the rupture occurred either on a left-lateral north–south striking fault or along a right-lateral east–west striking fault.

According to a press release issued by the Indonesia Meteorological, Climatological, and Geophysical Agency (BMKG) in the ASEAN Coordination Centre for Humanitarian Assistance on disaster management (AHA Centre), up to 27 October 2018 at 07.00 (UTC + 7), 770 aftershocks were recorded, with maximum and minimum magnitudes of M 6.3 and M 2.9, respectively [3]. Based on information obtained from the AHA Centre, aftershocks also triggered landslides in hilly areas due to the existing fault lines, while liquefaction also occurred in some areas, leading to loss of human life and damage to property [4]. For example, hundreds of people went missing from the Balaroa and Petobo villages, which were buried completely. 

The BMKG reported that a tsunami struck Talise Beach in the city of Palu, as well as beaches in Donggala [5]. Some settlements and buildings were damaged and several casualties occurred. The tsunami’s height was around 2.2–11.3 m, and it reached 0.5 km inland, destroying settlements and causing significant loss and massive damage. 

Indonesia’s BNPB reported that the total loss and damage caused by this disaster amounted to USD 95 million (13.82 trillion Rupiah); this estimate is expected to increase, as the release included only temporal data. The loss and damage caused by the disaster’s impact span five developmental sectors, including housing, infrastructure, the productive economy, the social sector, and cross-sector, with the greatest loss and damage occurring in the housing sector [1]. Up to 20 November 2018 at 18.00 UST + 7, the BNPB recorded 2101 fatalities and 68,451 damaged houses, with the greatest proportion of both categories for Central Sulawesi recorded in Palu [1]. 

Historically, this region has hosted several large earthquakes, with the largest event being an Mw 7.9 earthquake in January 1996 at Toli-Toli city [6], around 100 km north of the epicenter of this most recent event. The 1996 earthquake resulted in approximately ten fatalities and over 60 injuries, and caused significant building damage in the locality. According to an earlier study by Horspool et al. [7], who conducted a national probabilistic tsunami hazard assessment in Indonesia, which included Sulawesi Island among the study areas, over the last 100 years, the eastern part of Indonesia has experienced more massive tsunami events than has the western part. The northern arm of Sulawesi may generate earthquakes and tsunamis as a result of the subduction zone in the Sulawesi Sea [8]. Rahmadaningsi [8] constructed a simulation using a single fault method and simulated tsunami inundation using raster calculator functions in ArcGIS software, based on the 1996 earthquake that occurred at Toli-Toli, North Sulawesi. The result shows that bay-shaped shorelines are associated with higher tsunami run-ups than straight or cape-shaped shorelines. In addition to tsunamis, liquefaction has also occurred as a result of the intense shaking of earthquakes in some areas around Palu. However, relatively little research has focused on liquefaction in Indonesia [9]. Therefore, to adequately address and mitigate the damage caused by earthquakes, including tsunamis and liquefaction, a post-earthquake damage map is required as a first step in mitigating earthquake-associated hazards and in conducting risk studies. 

Two predominant methods are available for obtaining accurate information of post-earthquake damage, as mentioned by Xu [10]: field investigations and remote sensing image interpretation. Since remote sensing technology can provide an overview of damage sustained and is more cost-efficient than field investigations, this technology has become the more popular method for monitoring the aftermath of disastrous events [11,12,13,14,15,16,17]. Artificial intelligence is also a useful technology for assessing natural hazards, and has been used widely in disaster and risk management. For instance, recent studies have applied artificial intelligence, such as artificial neural networks (ANN) [18,19,20,21,22,23] and support vector machines (SVM) [24,25,26,27,28,29,30], to evaluate or analyze hazards in many areas. The application of remote sensing and artificial intelligence in many disastrous events has proven beneficial.

Therefore, the aim of the present study was to map post-earthquake damage in the city of Palu, Indonesia, using ANN, SVM, and decorrelation methods as a basis for risk assessments, prioritizing evacuation, and mitigation efforts in Palu.

## 2. Data and Methods

The earthquake occurred near the city of Palu, Sulawesi Island, Indonesia (Figure 1a). Palu is the capital of the Central Sulawesi Province, located at 0°53′42″S, 119°51′34″E (Figure 1b), and is characterized by various geological conditions. Sulawesi is a K-shaped island that can be divided into 14 geomorphic units [31]. Based on the scheme developed by Matsuoka et al. [32], Sulawesi’s geomorphological classes include the pre-tertiary mountains, tertiary mountains, hills, mountain foot slopes, volcanic foot slopes, volcano (which covers over 75% of the island’s area), and dunes. 

Sulawesi Island is located at the convergence of three plates: the Indian-Australian Plate, the Pacific Plate, and the Eurasian Plate [33]. Consequently, the island has several active faults, including the active Palu-Koro fault located in the northwest area of Palu (Figure 2) [34]. 

The Palu-Koro fault zone separates the Sulawesi Sea from the Makassar Basin, dividing it into the North Makassar and South Makassar Basins [36]. The Palu-Koro transform fault caused part of western Sulawesi to move in a south–southeasterly direction, which influenced the island’s position and caused two small spreading centers in the Makassar Strait to become dormant [36]. Further spreading accommodated by the left-lateral Palu-Koro transform fault included the northward movement of Sulawesi; as a result, Sulawesi moved farther from Kalimantan, which moved along a north-northwest–south-southeast axis [37]. Two main tectonic assemblages can be distinguished in Sulawesi, including the ophiolite complexes of the eastern and southern arms and the tertiary granites, volcanic, and recent and sub-recent volcanic deposits of the northern and southern arms [36]. Palu also has a unique geological structure that includes five main structures. As reported by Socquet et al. [38], there are four strands of the Palu-Koro strike–slip fault that cover an area of 50 km in width near Palu, where the slip range is 30–40 mm/year. The valley is composed of alluvial deposits; the northwestern part of the city consists of granite fragments, the western area in the northern region consists of alluvium and beach sediment rocks, while the southern region is composed of molasse sediment; and the east is composed of metamorphic rock (Figure 2).

Based on field data, the BMKG reported and shared photograph of the field conditions following the earthquake, as shown in Figure 3a–c. Figure 3a illustrates the conditions in Palu Bay, where the iconic Ponulele Bridge, which connected the east and west sub-districts of the city, collapsed; Figure 3b shows the area affected by the tsunami; and Figure 3c shows the region affected by liquefaction in Palu.

In this study, a post-earthquake damage map was generated using two artificial intelligence techniques, ANN and SVM, as well as a decorrelation method. Landsat-8 and Sentinel-2 data were used to ascertain the pre- and post-earthquake conditions. The collected Landsat and Sentinel images were classified using two artificial intelligence models and ANN and SVM classifiers. The pre- and post-earthquake classification results were then calculated using the decorrelation method, as detailed by the flow chart in Figure 4.

### 2.1. Satellite Imagery Data

Satellite imagery from Landsat-8 and Sentinel-2 were obtained for use in this study to generate the post-earthquake damage map of Palu, Central Sulawesi, Indonesia. Two images from Landsat-8, consisting of eight multispectral bands with a spatial resolution of 30 m, were selected for the pre-earthquake (Figure 5a) and post-earthquake (Figure 5b) conditions. 

The pre-earthquake image was captured on 16 September 2018, while the post-earthquake image was captured on 2 October 2018. Three multispectral bands were selected to generate the false color combination: shortwave near-infrared (SWIR2) (2.09–2.35 µm), thermal infrared (TIR) (10.40–12.50 µm), and near-infrared (NIR) (0.77–0.90 µm). A false color band combination was selected to easily differentiate the affected area from the area that unaffected by the earthquake during the classification stage.

Sentinel-2A and Sentinel-2B MSI level 1C data were captured for both pre- and post-earthquake conditions on 17 September 2018 (Figure 5c) and 22 October 2018 (Figure 5d), respectively. The multispectral bands used with a spatial resolution of 10 m were the three visible bands: NIR (842 nm), red (665 nm), and green (560 nm). Sentinel-2-A and 2-B differed in terms of the launch date: Sentinel-2A was launched on 23 June 2015, while Sentinel-2B was launched on 7 March 2017. Both images acquired by Sentinel-2 are at Level 1C, for which the per-pixel radiometric measurements are provided in top-of-the-atmosphere reflectance [40]. For this study, the Sentinel Application Platform toolbox was used to process the Sentinel-2 images (http://step.esa.int/main/toolboxes/snap/). Both the pre- and post-earthquake data from the Landsat and Sentinel imagery were covered by clouds in some regions, which could lead to misclassification and affect the final map result, particularly in the case of Sentinel images. However, cloud contamination in the Landsat image did not significantly cover the affected area, and both images were suitable for use.

### 2.2. Artificial Neural Network

ANN modeling is a reliable computational method for data classification that is applicable across several fields [41]. ANN is capable of acquiring, presenting, and calculating data for mapping [42]. The ANN model aims to form a method for predicting outputs from input data that are not used in the modeling process [43]. The back-propagation algorithm is the most frequently used neural network method and was used for this study. The back-propagation algorithm is trained using a set of examples of associated input and output values [21]. This learning algorithm is a multi-layered neural network that consists of input layers, a hidden layer, and an output layer (Figure 6). The input layers are the classes sampled from the satellite image processed by the hidden and output layer neurons by multiplying each input. At the end of the training phase, the neural network runs a model that should be able to predict a target value from a given input value [44]. This study referred to Kavzoglu’s [45] network architecture and suggested training patterns, wherein the training threshold contribution and momentum were set at 0.9. Meanwhile, the training rate was set at 0.2, the training root-mean-square exit criterion was set at 0.1, and the number of training iterations was 1000.

### 2.3. Support Vector Machine

The SVM algorithm is a useful classifier that is used widely for various types of classification because it works effectively with linearly non-separable and high-dimensional datasets [46,47,48]. SVM model performance is influenced by kernel functions, such as linear, radial basis, sigmoid, or polynomial functions [48,49]. The radial basis function (RBF) was selected for use in this study, due to its superior efficacy in nonlinear classification [50,51]. The formula and parameters for the RBF are as follows [52]:

K(x_i_,x_j_) = exp(−γ‖x_i_ − x_j_‖), γ > 0,
(1)
where K(x_i_,x_j_) denotes the kernel function, and γ is the gamma term in the kernel function for the RBF kernel, and should be optimized in the modeling process. This study employed the RBF as the neural network architecture for its superior association with input data [48,53], wherein the penalty parameter was set at 100, and γ in the kernel function was set at 0.333, to generate the most accurate model possible. 

### 2.4. Decorrelation Method

Decorrelation is a method whereby the discrimination between areas of different coherence is increased, and the areas are then termed decorrelated areas [54]. Decorrelation can also be defined as a method for generating decorrelating pixels in the image compression process [55]. Decorrelation terms are used in various fields, including statistics [55], signal processing [56], and image processing for synthetic-aperture radar (SAR) or optical images [57,58,59]. The decorrelation method used in this study was adapted from SAR data processing for earthquake damage detection, which has been referred to as an image-matching process [60] and similarity-based classification [61]. In this study, the method was basically adapted to differentiate two raster images using raster calculator analysis. Raster calculator analysis is useful for calculating the raster value differences between two raster images that overlap geographically [62]. Applying this analysis using ArcMap 10.4, the differences between the pre- and post-disaster images of Palu could be detected by subtracting the post-earthquake raster image from the pre-earthquake raster image. Consequently, a damage map was generated, showing the damaged area following the disaster’s occurrence. Similarity-based classification has been previously explained as follows: if the similarity is high, the building is likely to be still intact, while if the similarity is low, it is likely to have been destroyed [61]. Therefore, the output raster dataset from the subtraction arithmetic operator (Figure 7) was used to determine whether the value was 0 pixels, which would become a correlation result of two raster images and would not show the damage inflicted by the earthquake, or other pixel numbers (e.g., −2, −1, 1, and 2), which would show the decorrelation result and display the damaged area. Using this method, the post-earthquake damage map could be generated:
Expression: output raster = [Inlayer1] − [Inlayer2],
(2)


## 3. Results

### 3.1. Landsat Image Classification of ANN and SVM Classifiers

Landsat-8 images were used to classify the land-cover map of Palu before and after the disaster. ENVI Classic 5.3 software was used to process the classification, using both ANN and SVM models. Figure 8 shows the classification results of the ANN and SVM classifiers for the pre-disaster (Figure 8a,c) and post-disaster (Figure 8b,d) conditions. 

The classification map was divided into five classes (urban or affected area, vegetation, sea and river, bare land, and clouds) based on the colors shown by the false color band combination from the Landsat-8 satellite images. To generate the classification result, a stratified random sampling method on a pixel-by-pixel basis was used. Fifteen random training samples (polygons) were created in each class, yielding a total of 75 polygons or training samples for the Landsat image. In the resulting pre-earthquake map, red shading shows the urban area before having been affected by the earthquake, while in the post-earthquake map, red shading shows the urban area affected by the earthquake, including tsunami and liquefaction damage. In the post-earthquake image (Figure 8b,d), particularly to the south-east and center-east, two large affected areas, were detected by the ANN and SVM classifiers, where damage was caused by liquefaction. It was evident from the Landsat images in Figure 5a,b that liquefaction was the cause of changes in those areas. Since the decorrelation method was applied to nullify the same pixels and to show differences, it easier to identify the changes by applying the same color classification to the urban area in the pre-earthquake map and the affected area in the post-earthquake map. 

Figure 8c,d show the SVM classification results from the Landsat-8 image, which differed little from the ANN classification (Figure 8a,b). Following the same premise used to create the damage maps, the differences between the ANN and SVM classification results were further detected using the decorrelation method (see Section 3.5). 

In the pre- and post-earthquake SVM classification results, the urban or affected area classifications were indicated by red shading, as in the ANN classification. The urban area in the pre-earthquake image differed from that in the post-earthquake image, since the post-earthquake area was affected by the disastrous events associated with the earthquake, including the tsunami and liquefaction, in the densely populated area. Red shading was used in both SVM maps following the same premise as in the ANN classification: since the decorrelation method negated the same pixels, it was easier to generate the damage map.

### 3.2. Landsat Images: Decorrelation of the ANN and SVM Classifiers

The results of the ANN and SVM classifications from the Landsat-8 images were analyzed using the raster calculator function in ArcMap 10.4 software. Using the raster calculator to subtract the post-earthquake image data from that of the pre-earthquake image, differentiation maps of pre- and post-earthquake images from the ANN and SVM classifications were generated, which could be considered as post-earthquake damage maps (Figure 9).

The ANN decorrelation results (Figure 9a,b) showed that the area affected by the earthquake extended from the coastal line to the southern part of the city, due to the tsunami and liquefaction that occurred in some areas after the earthquake. The coastal areas near Palu Bay were damaged by the earthquake and tsunami, and houses in the city were affected by the earthquake. Three villages, namely Petobo, Balaroa, and—the largest—Sidera–Jono Oge, were also affected by earthquake-triggered liquefaction. The total affected area of the damage map based on the ANN classification was calculated by multiplying the total number of pixels by the cell size of the Landsat image, which yielded an area of 8,814,600 m^2^ or 881.46 ha. This result did not differ substantially from the total damaged area based on field data reported by the BNPB of 390.82 ha for the liquefaction area and 2976 houses; although there was no mention of the total area, the total damage to Palu could be assumed to have reached 850 ha. However, the results in Figure 9a,b erroneously indicated some areas as having been affected due to differences in cloud positioning between the pre- and post-disaster images.

The SVM decorrelation results are shown in Figure 9c,d, where red shading indicates differences between the pre- and post-earthquake maps, categorized as the areas affected by the disaster. The coastal areas incurred heavy damage because of the tsunami, while large inland mud flows that caused severe damage in densely populated areas were caused by earthquake-triggered liquefaction in the three aforementioned villages. Based on the SVM decorrelation process, the total damaged area was calculated as 7,974,000 m^2^ or 797.4 ha, which did not differ substantially from the total damaged area indicated by the field data. As with the ANN-derived results, some areas were erroneously categorized as damaged due to differences in cloud cover in the pre- or post-earthquake Landsat images.

The generated map yielded similar results to the ANN decorrelation, where some areas were found to have changed as a result of the earthquake, tsunami, and liquefaction. Since the results based on SVM decorrelation were very similar to those based on ANN decorrelation, a post-earthquake decorrelation map was generated from the two post-earthquake classification methods (ANN and SVM) to differentiate the damaged area from the two classifiers (see Section 3.5).

### 3.3. Sentinel Image Classification using ANN and SVM Classifiers

Sentinel-2 false color combination images were used for classification with the ANN and SVM methods. The pre- and post-earthquake images were categorized into six classes: urban or affected area, sea and river, vegetation, bare land, clouds, and shadows (Figure 10). To produce these images, same sampling methods and training sample distribution were applied to the Landsat image, which was stratified using a random sampling method on a pixel-by-pixel basis. After the classes had been defined, 15 random training samples (polygons) were created in each class, yielding 85 polygons in total for the Sentinel images. The purpose of using red to indicate the urban or affected area in the pre- and post-earthquake Sentinel images was similar to that in the case of the Landsat images: since the decorrelation subtracts the pre- and post-earthquake images to show the differences in pixels, it is simpler if the post-earthquake affected area is indicated in the same color as the urban area before the earthquake’s occurrence. However, some areas classified by the ANN were determined as another class: for example, in the pre-earthquake map, shadows were classified as water in some areas, while clouds were classified as urban area in the southwest part of the map. Meanwhile, urban area was classified as blue in the post-earthquake image since the color was similar to water in the sea and river class. In the post-earthquake classification map, some vegetation in the northeastern areas was also classified as bare land, indicated as yellow shading. Finally, differences in the positioning of clouds in the pre- and post-earthquake images influenced the damage map generated using the decorrelation method (see Section 3.4). 

The SVM classification results for the Sentinel-2 images are shown in Figure 10c,d. As described above, both urban and affected areas were indicated in red shading for easier generation of the post-earthquake damage map using the decorrelation method. However, the pre-earthquake (Figure 10c) and post-earthquake (Figure 10d) classification images differed somewhat, particularly in the urban area and shadow classes. In the pre-earthquake classification result, some urban areas were classified as cloud (cyan shading), while in the post-earthquake classification result, some shadows were classified as water (blue shading), and some clouds were classified as urban area, particularly in the case of a cloud that appeared in the center of the map. Although this may have influenced the decorrelation results, the main focus was to assess the post-earthquake damage in the urban area as classified by the SVM.

### 3.4. Sentinel Image Decorrelation of ANN and SVM Classifiers

The ANN and SVM classification results from the Sentinel-2 imagery were analyzed using the decorrelation method. The pre- and post-earthquake classifications were subtracted to generate damage area maps (Figure 11). 

The post-earthquake damage maps from the ANN and SVM classifications showed the areas affected by the earthquake, indicated in red. However, in the results, differences in cloud position were also categorized as damaged area, which ultimately dominated the post-earthquake damage maps. The total area indicated in red in the ANN classification was 11,502,100 m^2^ or 11.150 ha; this was much greater than the damaged area estimated from the field data due to clouds that dominated the map. Both the pre- and post-earthquake Sentinel-2 images were cloudy, causing some areas to be detected as damaged area, and influencing the map’s accuracy in terms of its comparability to the field data.

Figure 11c,d show the decorrelation results for the SVM classification of the Sentinel-2 images in true color and hillshade view, respectively. In the northeast part of the map, clouds were identified as damaged area due to the differences in cloud positions in the pre- and post-earthquake damage maps. The total damage area from the SVM classification method was 31,941,100 m^2^ or 31.941 ha, due to the inclusion of clouds that dominated the map. This area differed substantially from area estimated from the field data, due to the cloudy conditions.

### 3.5. Conformity of the ANN- and SVM-derived Damage Maps based on Landsat and Sentinel Images

Decorrelation maps for differentiating the results of the SVM and ANN decorrelation maps were made using the post-earthquake results from the Landsat and Sentinel images (Figure 12). The maps revealed some differences in classifications between the two methods (blue and yellow shading, Figure 12). 

From the Landsat images (Figure 12a), areas classified only by the ANN method (blue shading) occupied a greater area than did areas only classified by the SVM method (yellow shading). This indicated that the SVM results only differed slightly in the classification of the post-earthquake map compared to the ANN results in terms of the total area for each post-earthquake damage map. The ANN decorrelation map showed a greater area of damage than did the SVM map, (ANN: 881.46 ha, SVM: 797.4 ha). However, both classifiers yielded very similar results, as calculated using the following conformity equation:

conformity = 1 − ((A + B))/((A + B + C)) × 100%,
(3)
where A is the total number of pixels only in the ANN damage map, B is the total number of pixels only in the SVM damage map, and C is the intersection of the total pixels in the ANN and SVM damage maps. When A = 1136 pixels, B = 286 pixels, and C = 8614 pixels, the conformity was 85.83%, indicating high conformity between the ANN and SVM decorrelations for the Landsat image map.

The same process was performed for the Sentinel-2 decorrelation maps to ascertain the differences between the ANN and SVM decorrelation pre- and post-disaster map results (Figure 12b). To distinguish them, a raster calculator analysis was applied to subtract the raster map from the post-disaster ANN and SVM classification results. After subtracting the images, the distribution of damaged area only in the SVM classification was indicated in yellow, whereas that only in the ANN classification was indicated in blue. As demonstrated in Figure 12b the ANN results from the Sentinel-2 image occupied a smaller area than did the SVM results. The SVM method classified several areas differently compared to the ANN method, including in the western part of the map, where clouds were classified as damaged area using the SVM method. The conformity result was calculated using Equation (3). When A = 11,202 pixels, B = 145,682 pixels, and C = 277,458 pixels, the percentage conformity was 63.88%, meaning that the conformity between the ANN- and SVM-based decorrelation map in Figure 12b was weak.

## 4. Discussion

Classification maps using ANN and SVM classifiers generated from Landsat-8 and Sentinel-2 images were produced. Overall, there were eight pre- and post-earthquake classification maps, wherein all Landsat-8 imagery was classified into five classes, urban area or afected area, vegetation, sea and river, bare land, and clouds, while the Sentinel-2 imagery was classified into six classes, with the addition of shadow as the sixth class. There were some differences among the image classification results that were indirectly influenced by the image source and sampling methods used to determine the area of each class. These classification results were then analyzed by decorrelation to reveal the differences between the pre- and post-earthquake images. From the eight resulting classification maps, four decorrelation maps were generated, divided by classifier type (ANN and SVM classifiers) and image source (Landsat-8 and Sentinel-2 imagery). The decorrelation maps, which also included the results of the post-earthquake damage maps, were compared to the field data. For the Landsat-8 images, the results and total area were very similar, but for the Sentinel-2 images, the damaged areas were dominated by clouds rather than the actual damaged area, due to the image conditions. In this case, Landsat-8 yielded superior results for the post-earthquake damage map than did the Sentinel-2 images.

The BNPB, Geospatial Information Agency, and National Institute of Aeronautics and Space of Indonesia have published field data, including distribution maps of the areas affected by the tsunami, earthquake, and liquefaction. In comparison to the field data map (Figure 13), the results yielded by the ANN and SVM methods, along with decorrelation, effectively mapped the damaged area, particularly in the case of the Landsat-8 results. As may be seen from Figure 13a, the damage to coastal areas caused by the tsunami and earthquake were the same as those in the decorrelation maps of the SVM and ANN classifiers. 

Furthermore, Figure 13b shows that the region affected by liquefaction also confirmed that the ANN and SVM classifier decorrelation method was effective in establishing the damaged area resulting from liquefaction, particularly in the Petobo, Sidera, and Jono Oge villages. Based on data released by the BNPB, a total of 68,451 buildings were damaged, of which 2976 were damaged by the tsunami. The highest concentration of damage occurred in Lere Village, in West Palu Sub-District, with 786 buildings affected. Most of the damage to houses and buildings was caused by liquefaction, with a total of 496 units damaged in Jono Oge and Sidera, 1357 units damaged in Balaroa, and 1920 units in Petobo. In addition to houses, essential facilities such as schools, pharmacies, government offices, and hospitals were also damaged by this disaster. For this reason, post-earthquake damage mapping is a useful and effective means of developing early evacuation and mitigation strategies in affected areas, such as Palu, Central Sulawesi, Indonesia, which are not easily accessed. 

## 5. Conclusions

Post-earthquake damage maps of the city of Palu, where the 2018 Mw 7.4 earthquake occurred, was generated. Two satellite image types were employed, Landsat-8 and Sentinel-2 images, and both satellite images were classified using ANN and SVM methods. The decorrelation method was used to generate damage maps from the pre- and post-earthquake ANN and SVM classification maps. The decorrelation map results were assessed to ascertain the correlations between the ANN and SVM methods, which yielded very similar results. Moreover, the analysis and verification results indicate satisfactory agreement between the post-earthquake damage maps and field data. 

Post-earthquake damage maps are an essential tool in post-earthquake hazard mapping. Therefore, the results of this study are expected to prove particularly useful in high-risk scenarios, such as that in Palu, Central Sulawesi, and throughout Indonesia, as a country with many active faults. The developed post-earthquake damage maps are beneficial to planners, engineers, and governments, and can help develop evacuation or mitigation strategies to minimize the number of victims, and determining the appropriate courses of action for further plans and developments. However, while the methods used in this study are valid for evacuation and mitigation purposes in areas with a history of disastrous occurrences, they are likely to be less useful for sites without any disastrous events on record. Further research should prioritize the application of post-disaster damage maps to other regions, particularly in Indonesia, which is highly susceptible to various natural hazards, including earthquakes, landslides, subsidence, tsunamis, and flooding.

## Figures and Tables

**Figure 1 sensors-19-00542-f001:**
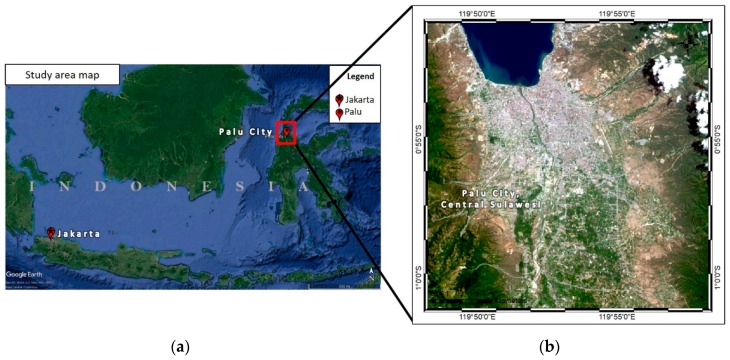
Map of Palu, Central Sulawesi Province, Indonesia, where the highest level of damage occurred as a result of the 2018 earthquake: (**a**) location of Palu in relation to Jakarta, the capital of Indonesia; (**b**) image of Palu from Landsat-8 data captured on 16 September 2018.

**Figure 2 sensors-19-00542-f002:**
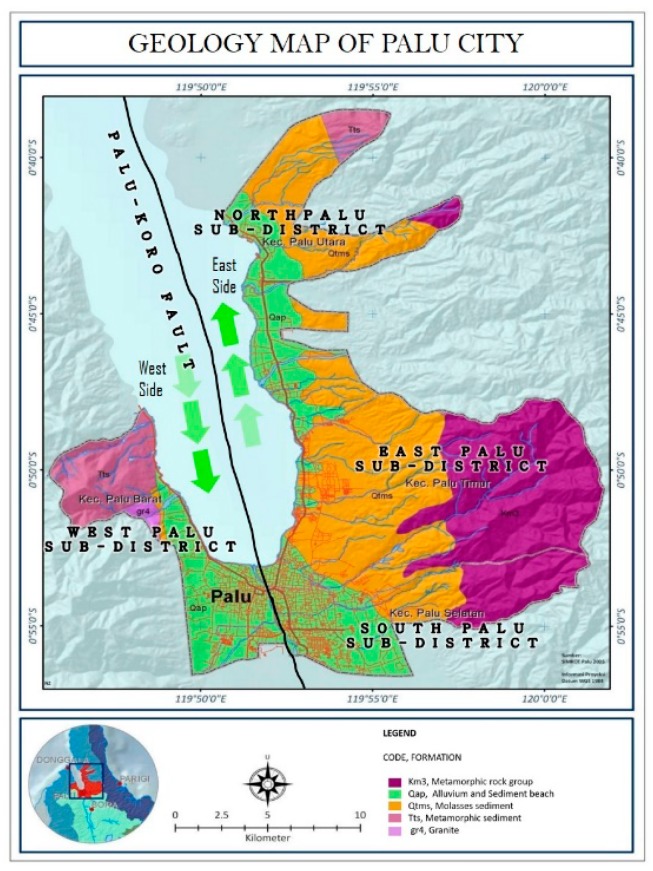
Geological map of Palu, Indonesia, which has five main geological structures: the metamorphic group, alluvium and sediment beach group, molasse sediment, metamorphic sediment, and granite. The map also shows the Palu-Koro fault, which caused the earthquake in Palu. Modified from [35].

**Figure 3 sensors-19-00542-f003:**
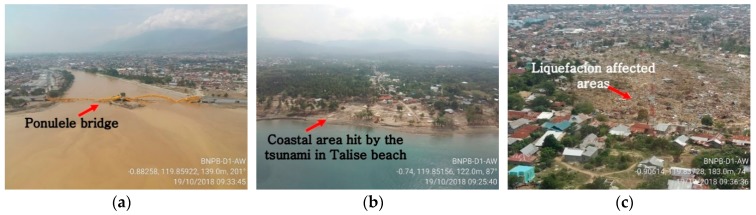
Photographs of the conditions after the Mw 7.4 earthquake in Palu, Indonesia: (**a**) collapse of the iconic Ponulele Bridge; (**b**) tsunami damage in the coastal area; and (**c**) liquefaction within an area of Palu. Modified from [39].

**Figure 4 sensors-19-00542-f004:**
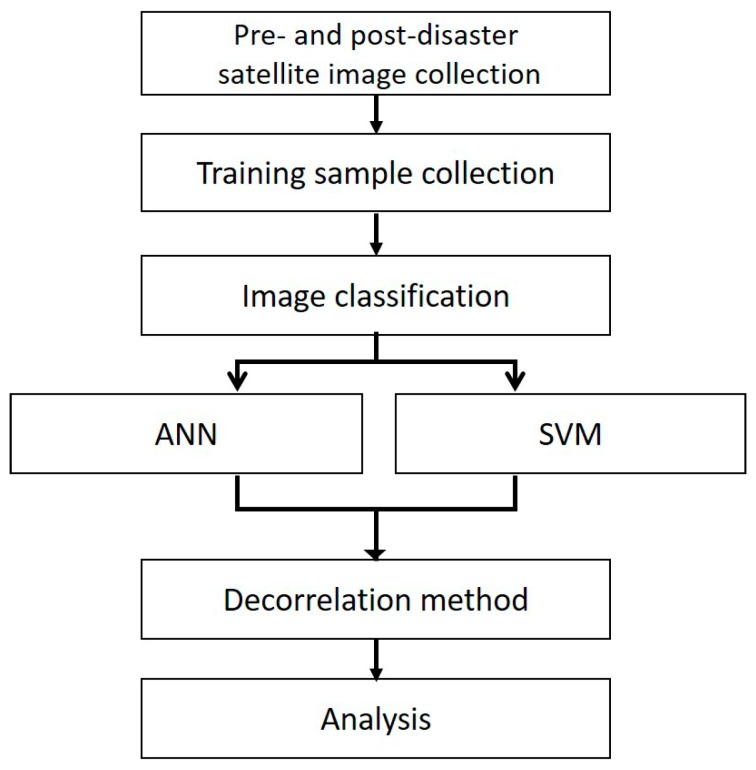
Flow chart of the method used to generate the damage maps for the Palu earthquake disaster.

**Figure 5 sensors-19-00542-f005:**
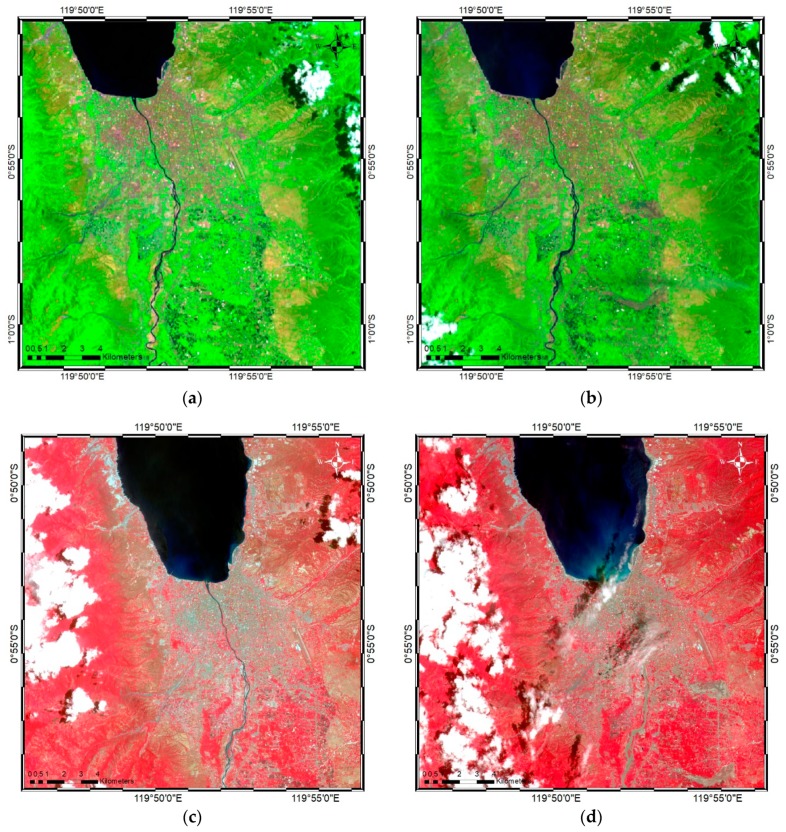
False color combination images of Palu, Indonesia, captured by Landsat-8 (**a**) pre-earthquake on 16 September 2018 and (**b**) post-earthquake on 2 October 2018, and by Sentinel-2 on (**c**) 17 September 2018 and (**d**) 22 October 2018.

**Figure 6 sensors-19-00542-f006:**
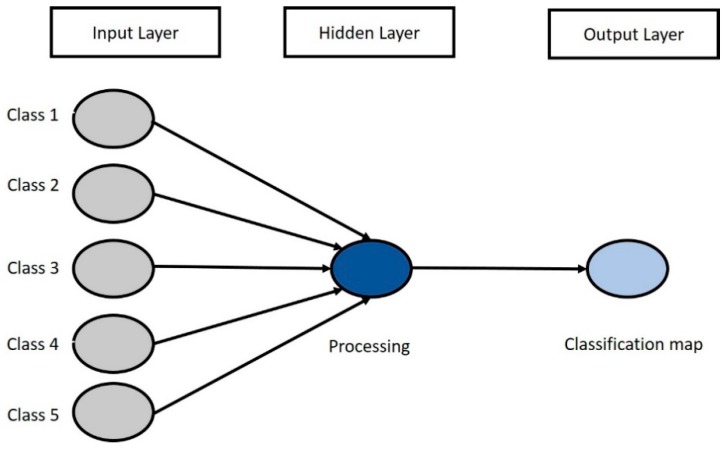
The artificial neural network model used to generate the classification map. Classes 1–5 are the input layers from satellite images, which were selected using ENVI software.

**Figure 7 sensors-19-00542-f007:**
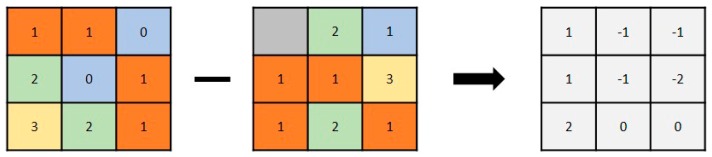
Example of the arithmetic operation used for two raster images to generate the post-earthquake damage map of Palu, Indonesia.

**Figure 8 sensors-19-00542-f008:**
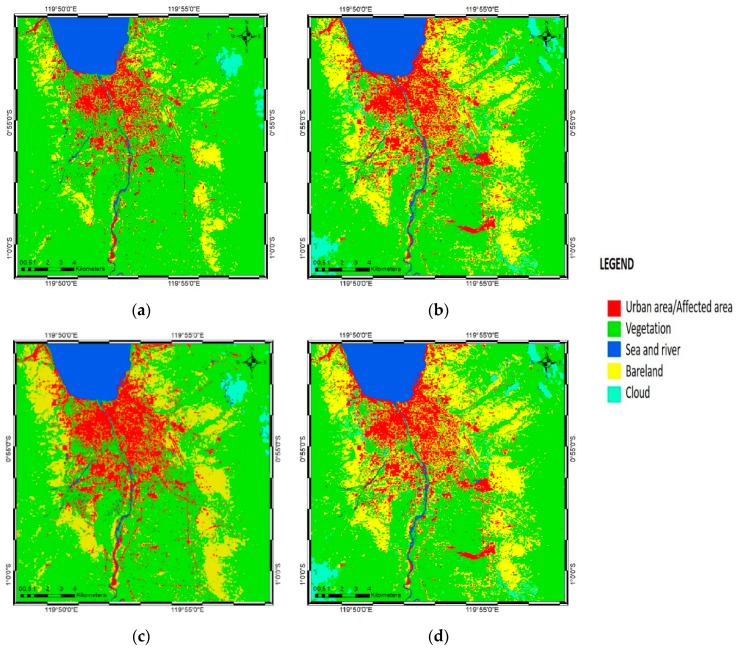
Results of the ANN and SVM classifications based on Landsat-8 imagery of Palu, Indonesia: (**a**) ANN pre-earthquake map; (**b**) ANN post-earthquake map; (**c**) SVM pre-earthquake map; and (**d**) SVM post-earthquake map.

**Figure 9 sensors-19-00542-f009:**
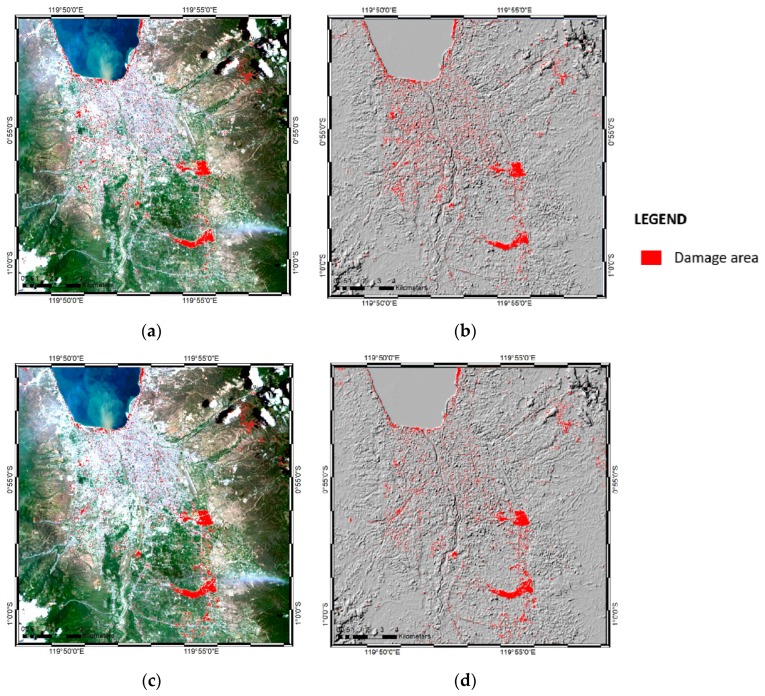
Post-earthquake damage maps of Palu, Indonesia, generated using the decorrelation method from the (**a**) ANN classification results in true color view, (**b**) ANN classification results in hillshade view, (**c**) SVM classification results in true color view, and (**d**) SVM classification results in hillshade view.

**Figure 10 sensors-19-00542-f010:**
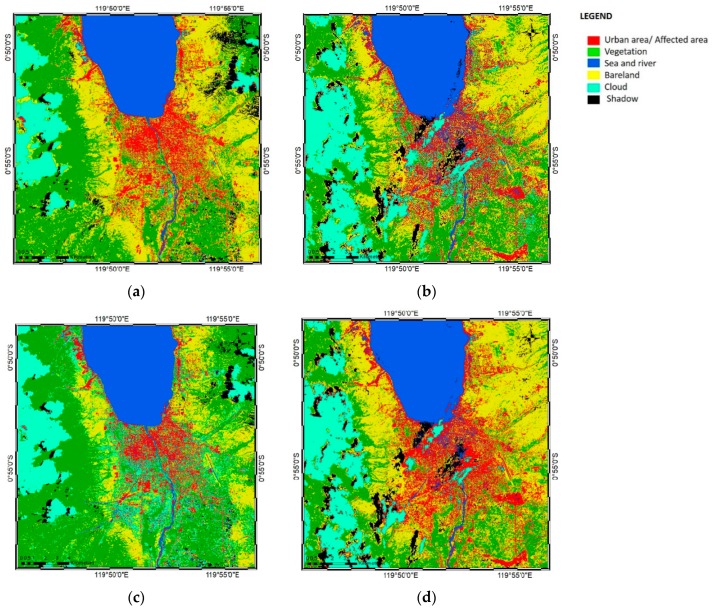
Sentinel-2 classification image results of (**a**) ANN classifier pre-earthquake occurrence, (**b**) ANN classifier post-earthquake occurrence, (**c**) SVM classifier pre-earthquake occurrence, and (**d**) SVM classifier post-earthquake occurrence in Palu, Indonesia.

**Figure 11 sensors-19-00542-f011:**
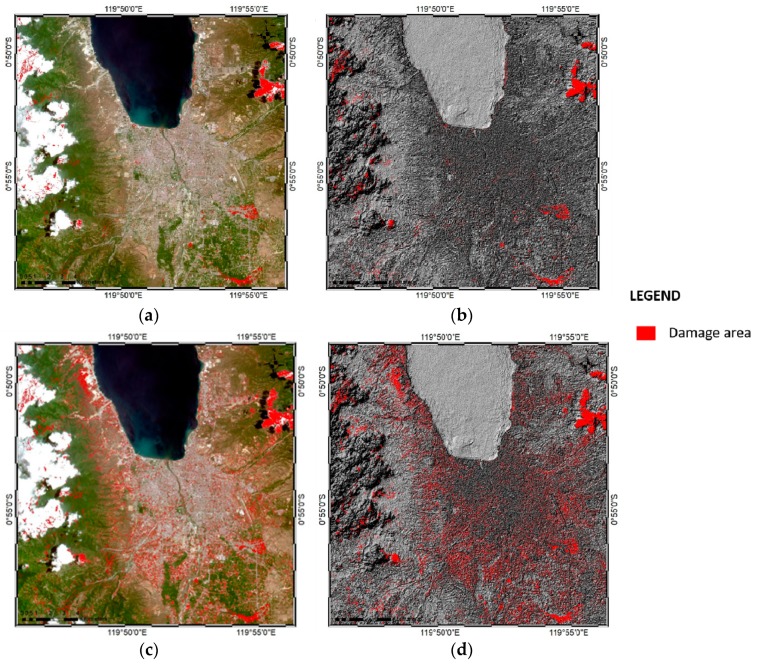
Post-earthquake damage maps of Palu, Indonesia, generated using the decorrelation method from the ANN and SVM classification results of the Sentinel-2 images in (**a**) ANN true color view, (**b**) ANN hillshade view, (**c**) SVM true color view, and (**d**) SVM hillshade view.

**Figure 12 sensors-19-00542-f012:**
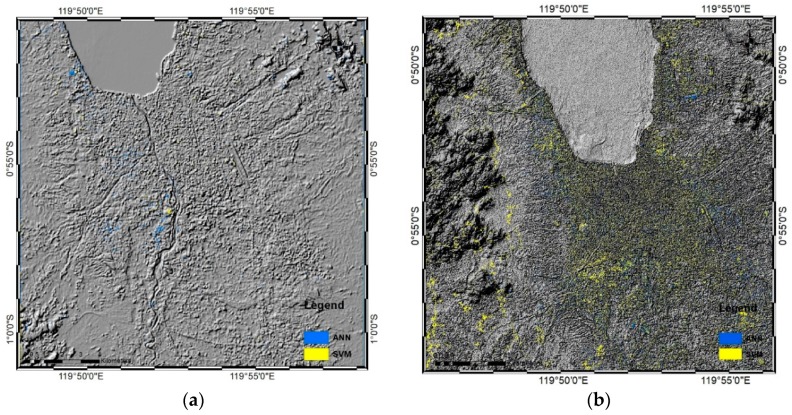
Decorrelation maps showing the differences between (**a**) the Landsat-8 ANN and SVM results for the post-earthquake damage area and (**b**) the Sentinel-2 ANN and SVM results for the post-earthquake damage area in hillshade view.

**Figure 13 sensors-19-00542-f013:**
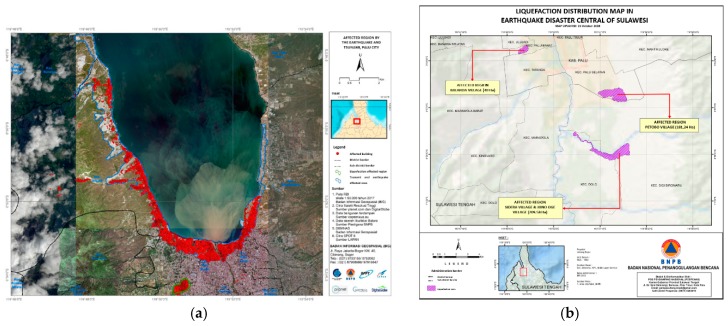
Field data from the National Board for Disaster Management of Indonesia: (**a**) the region affected by the earthquake and tsunami in Palu, Indonesia, on 28 September 2018; (**b**) liquefaction distribution map. Modified from [1].

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
