# Peer review of "An Artificial Intelligence Application for Post-Earthquake Damage Mapping in Palu, Central Sulawesi, Indonesia"

_sensors, 2019, doi:10.3390/s19030542_

Round 1

Reviewer 1 Report

In 2018, the Indonesian Palu earthquake triggered a large number of landslides, liquefaction, and deadly tsunami. The article uses Landsat and Sentinel data, ANN and SVM models to carry out the mapping research of post-earthquake disasters, which has positive significance for understanding the seismic damage distribution of the Palu earthquake and the later earthquake disaster reduction. The specific issues are as follows:

(1) How does the author choose the SVM and ANN model training samples? And give the distribution of the samples.

(2) The resolution of Sentinel data is better than Landsat. However, the result is that Landsat is much better than Sentinel data. What is the reason?

(3) Please give direct evidence or images of these affected areas in Figure 20. Because of based on the high-resolution image, the Affected region Sibalaya Village does not seem to produce liquefaction.

(4) Be 2.101 should 2,101?

Author Response

Reviewer 1

Open Review

English language and style

( ) Extensive editing of English language and style required 
(x) Moderate English changes required 
( ) English language and style are fine/minor spell check required 
( ) I don't feel qualified to judge about the English language and style 

Yes

Can be improved

Must be improved

Not applicable

Does the introduction provide sufficient background and include   all relevant references?

(x)

( )

( )

( )

Is the research design appropriate?

(x)

( )

( )

( )

Are the methods adequately described?

(x)

( )

( )

( )

Are the results clearly presented?

( )

(x)

( )

( )

Are the conclusions supported by the results?

(x)

( )

( )

( )

Comments and Suggestions for Authors

In 2018, the Indonesian Palu earthquake triggered a large number of landslides, liquefaction, and deadly tsunami. The article uses Landsat and Sentinel data, ANN and SVM models to carry out the mapping research of post-earthquake disasters, which has positive significance for understanding the seismic damage distribution of the Palu earthquake and the later earthquake disaster reduction. The specific issues are as follows:

How does the author choose the SVM and ANN model training samples? And give the distribution of the samples.

Answer: Thank you for the good question. The authors used stratified random sampling method on pixel by pixel basis. The classes were divided into five: urban area or affected area, vegetation, sea and river, bare land, and cloud for the Landsat images and six classes for Sentinel images namely:  urban area or affected area, vegetation, sea and river, bare land, cloud, and shadow.  After determining the classes, 15 random training samples (polygons) were created in each class, so there are 75 polygons for Landsat images and 85 polygons for Sentinel images. 

The author add this explanation in classification result:

“To generate the classification result, a stratified random sampling method on a pixel-by-pixel basis was used. Fifteen random training samples (polygons) were created in each class, yielding a total of 75 polygons or training samples for the Landsat image.” (Line 225-227)

“To produce these images, same sampling methods and training sample distribution were applied to the Sentinel image, which was stratified using a random sampling method on a pixel-by-pixel basis. After the classes had been defined, 15 random training samples (polygons) were created in each class, yielding 85 polygons in total for the Sentinel images..” (Line 289-293)

The resolution of Sentinel data is better than Landsat. However, the result is that Landsat is much better than Sentinel data. What is the reason?

Answer: Thank you for your question. In this study, the Landsat performs better result due to the clear image condition than the Sentinel data. Cloudy condition in Sentinel data affected the result of decorrelation map which some clouds determined as post-earthquake affected area or damage area. The author address this problem in result section which written as:

“…, differences in the positioning of clouds in the pre- and post-earthquake images influenced the damage map generated using the decorrelation method…” (Line 302-304)

“The post-earthquake damage maps from the ANN and SVM classifications showed the areas affected by the earthquake, indicated in red. However, in the results, differences in cloud position were also categorized as damaged area, which ultimately dominated the post-earthquake damage maps... Both the pre- and post-earthquake Sentinel-2 images were cloudy, causing some areas to be detected as damaged area, and influencing the map’s accuracy in terms of its comparability to the field data.” (Line 327-334).

Please give direct evidence or images of these affected areas in Figure 20. Because of based on the high-resolution image, the affected region Sibalaya Village does not seem to produce liquefaction.

Answer: Thank you for the questions. Figure 20 which become Figure 13b is the liquefaction affected areas that produced by National Board for Disaster Management (BNPB) of Indonesia. However the high-resolution image was limited to show all affected area due to the earthquake such as coastal and urban area in one frame image, which makes Sibalaya Village does not appear in high resolution image. In consequence the author solve this problem through editing Figure 20 as can be seen in Figure 13b (Line 395).

Be 2.101 should 2,101?

Answer: Thank you for the correction. Yes, it should be 2,101 fatalities. The author modified it in line 6.

Submission Date

11 December 2018

Date of this review

13 Dec 2018 04:16:19

Reviewer 2 Report

The manuscript applies Artificial Intelligence to the analysis of remote sensing imagery in order to map the damage due to the 2018 M7.9 Palu (Indonesia) earthquake. English in this manuscript is a quite significant problem. Honestly, I don't quite understand about this manuscript, largely because of the non-standard English. 

Author Response

Reviewer 2

Open Review

English language and style

(x) Extensive editing of English language and style required 
( ) Moderate English changes required 
( ) English language and style are fine/minor spell check required 
( ) I don't feel qualified to judge about the English language and style 

Yes

Can be improved

Must be improved

Not applicable

Does the introduction provide sufficient background and include   all relevant references?

( )

( )

(x)

( )

Is the research design appropriate?

( )

( )

(x)

( )

Are the methods adequately described?

( )

( )

(x)

( )

Are the results clearly presented?

( )

( )

(x)

( )

Are the conclusions supported by the results?

( )

( )

(x)

( )

Comments and Suggestions for Authors

The manuscript applies Artificial Intelligence to the analysis of remote sensing imagery in order to map the damage due to the 2018 M7.9 Palu (Indonesia) earthquake. English in this manuscript is a quite significant problem. Honestly, I don't quite understand about this manuscript, largely because of the non-standard English. 

Answer: Thank you for your comments and suggestions, the author will address this problem through English checking service.

Submission Date

11 December 2018

Date of this review

20 Dec 2018 23:39:28

Reviewer 3 Report

General commends:

The English language needs extended correction.

In many cases singular voice has been used instead of the correct plural one.

M--> Mw

meters --> m.

Use the term image and imagery with more caution

Please don't make use of third and fourth levels of section numbering.

Use a common mask applied to pre and post images of the total clouds and shadow. At least for the same sensors. This way the results will be more robust. 

The rationalism of the decorrelation method is very poor. In each one of the two datasets a third image should be added before or after the earthquake and be compared with the one before or after the earthquake, respectively, to persuade for the robustness of the methodology. See details below.

Specific comments:

10: M --> Mw

11: 2.101 --> 2,101

14: Pre- and post-earthquake images from 14 satellite imagery --> Pre- and post-earthquake satellite images

16: Further ...map. : Please rephrase

19: ...gives a better results... --> performs better...

20: ...with...classifiers... : please rephrase

21: Further...hazards : Please rephrase

I went through the abstract for English and style checking but since the edits needed are not spontaneous but the rule I didn't process it further. Thus from point forward I will comment only for the scientific part.

80 --> Material and methods --> Data and Methods

92-99 : Please show these faults in the map

Section 2.4. The decorrelation Method, even if it technically over explained, is purely supported by rational evidence and references in order to be used as actually change detection methodology which results to a post-earthquake damage map. It is not persuasive why a subtraction of classes numbers lead to the damage map.Q This section needs reduction of the technical details and expansion of the rationalism hidden within, not forgetting to add references.

Figures 4,5,8, 9, 10, 11, 12, 13, 14, 15, 16  and 18: please keep the map extension in all the figures unless there is a special reason, but please don't mix different map extensions during a processing chain or when compare products side but side like figure 9! Please stay in one or two map extension(s) which you should mark in Figure 2. Please reduce the number of the figures by merging the maps of the same product chain, so the classifications and correlations products to be in the same figure.

In Fig. 8b, 9b, 10b and 11b, in the SE and center-East (named in section 4 as Jono Oge and Sidera Villages and Petobo village, respectively ) there are two large areas which doesn't exist in (a). It seems to the reader that these urban areas exist on October but not in September. Please resolve this.

Author Response

Reviewer 3

Open Review

English language and style

(x) Extensive editing of English language and style required 
( ) Moderate English changes required 
( ) English language and style are fine/minor spell check required 
( ) I don't feel qualified to judge about the English language and style 

Yes

Can be improved

Must be improved

Not applicable

Does the introduction provide sufficient background and include   all relevant references?

(x)

( )

( )

( )

Is the research design appropriate?

( )

( )

(x)

( )

Are the methods adequately described?

( )

( )

(x)

( )

Are the results clearly presented?

( )

( )

(x)

( )

Are the conclusions supported by the results?

( )

( )

(x)

( )

Comments and Suggestions for Authors

General commends:

The English language needs extended correction.

In many cases singular voice has been used instead of the correct plural one.

Answer: Thank you for the comment and correction. The author will fix the English problem in this manuscript by submitting to English editing service.

M--> Mw

Answer: Thank you for the correction. The author change the M to be Mw; line 5, 21, 48, 117, and 412.

meters --> m.

Answer: Thank you for the correction. The author change meters to be m; line 39 and 134,

Use the term image and imagery with more caution

Answer: Thank you for the correction. The author change imagery term to be image; line 165.

Please don't make use of third and fourth levels of section numbering.

Answer: Thank you for the comment. The author reduce the section especially in the result part to two section numbering by replace some related result. For example the classification result for ANN and SVM from Landsat or Sentinel Images were merge in one section which also reduce number of figures; line 214- 247, line 286-318 Also for the decorrelation result for ANN and SVM from Landsat or Sentinel images were merge in one section; 248-285, line 320-340.

Use a common mask applied to pre and post images of the total clouds and shadow. At least for the same sensors. This way the results will be more robust. 

Answer: Thank you for the suggestion. The author follow this suggestion by ordering the images of pre- and post- based on the same sensor or same product chain Figure 5, 8, 9, 10, 11, and 12.

The rationalism of the decorrelation method is very poor. In each one of the two datasets a third image should be added before or after the earthquake and be compared with the one before or after the earthquake, respectively, to persuade for the robustness of the methodology. See details below.

 Specific comments:

10: M --> Mw

Answer: Thank you for the correction. The author change the M to be Mw; line 5, 21, 48, 117, and 412.

11: 2.101 --> 2,101

Answer: Thank you for the correction. Yes, it should be 2,101 fatalities. The author modified it in line 6.

14: Pre- and post-earthquake images from 14 satellite imagery --> Pre- and post-earthquake satellite images

Answer: Thank you for the correction, the author modified the sentence in line 14 to be:

“Pre- and post-earthquake satellite images..” in line 10.

16: Further ...map. : Please rephrase

Answer: Thank you for the comment. The author rephrase the sentence to be:

“The affected areas were compared to the field data, the percentage conformity between the ANN and SVM results was analyzed, and four post-earthquake damage maps were generated.” (Line 12-16)

19: …gives a better results… à performs better…

Answer: Thank you for the comment. The author modified the sentence after having an English correction to be:

“…was superior to that of Sentinel-2 (63.88%).” (Line 15)

20: ...with...classifiers... : please rephrase

Answer: Thank you for the comment. The author rephrase the sentence to be:

“Based on the conformity analysis, the Landsat-8 imagery (85.83%) was superior to that of Sentinel-2 (63.88%).” (Line 14-15)

21: Further...hazards : Please rephrase

Answer: Thank you for the comment. The author rephrase the sentence to be:

“The resulting post-earthquake damage map can be used to assess the distribution of seismic damage following the Palu earthquake and may be used to mitigate damage in the event of future earthquakes.” (Line 15-17)

I went through the abstract for English and style checking but since the edits needed are not spontaneous but the rule I didn't process it further. Thus from point forward I will comment only for the scientific part.

80 --> Material and methods --> Data and Methods

 Answer: Thank you for the correction. The author modified the subtitle to be:

“Data and Methods” (line 77)

92-99 : Please show these faults in the map

Answer: Thank you for your comment. The author modified Figure 2 and shows the Palu Koro faults which was described in 92-99. Figure 2 can be seen in line 108.

Section 2.4. The decorrelation Method, even if it technically over explained, is purely supported by rational evidence and references in order to be used as actually change detection methodology which results to a post-earthquake damage map. It is not persuasive why a subtraction of classes numbers lead to the damage map. This section needs reduction of the technical details and expansion of the rationalism hidden within, not forgetting to add references.

 Answer: Thank you for your comment and suggestion. The author edited some sentences and references to explain the decorrelation method in line 189-212.

Figures 4,5,8, 9, 10, 11, 12, 13, 14, 15, 16  and 18: please keep the map extension in all the figures unless there is a special reason, but please don't mix different map extensions during a processing chain or when compare products side but side like figure 9! Please stay in one or two map extension(s) which you should mark in Figure 2. Please reduce the number of the figures by merging the maps of the same product chain, so the classifications and correlations products to be in the same figure.

Answer: Thank you for your comments. The author edited some figures such as figure 2, which is separated to be figure 2 and figure 3. (Line 108 and 117). Also merged some figure that has same product chain such as ANN and SVM classification results from Landsat were merged to be in one Figure (Figure 8, line 220), or decorrelation result which also merged in one figure (Figure 9, line 255). By merging the same product chain, section number and figure number were reduce.

In Fig. 8b, 9b, 10b and 11b, in the SE and center-East (named in section 4 as Jono Oge and Sidera Villages and Petobo village, respectively ) there are two large areas which doesn't exist in (a). It seems to the reader that these urban areas exist on October but not in September. Please resolve this.

 Answer: Thank you for the comments and suggestions. The author is already address the urban area which marked by red color in figure 8b, 9b, 10b, and 11b which become 8b, 8d, 10b, and 10b in line 220 and 306, but to make it clearer the author write change the sentences to explain the area that seems to be appear suddenly in Figures 8b, 8d, 10b, and 10b to be:

“In the resulting pre-earthquake map, red shading shows the urban area before having been affected by the earthquake, while in the post-earthquake map, red shading shows the urban area affected by the earthquake, including tsunami and liquefaction damage. In the post-earthquake image (Figures 8b and 8d), particularly to the south-east and center-east, two large affected areas, were detected by the ANN and SVM classifiers, where damage was caused by liquefaction. It was evident from the Landsat images in Figures 5a and 5b that liquefaction was the cause of changes in those areas. Since the decorrelation method was applied to nullify the same pixels and to show differences, it easier to identify the changes by applying the same color classification to the urban area in the pre-earthquake map and the affected area in the post-earthquake map. “(Line 227-236)

Also for the figure 10b and 10d, the author refers this paragraph to explain the urban area/ affected area means, so the reader will understand the reason behind it as written to be: “The purpose of using red to indicate the urban or affected area in the pre- and post-earthquake Sentinel images was similar to that in the case of the Landsat images: since the decorrelation subtracts the pre- and post-earthquake images to show the differences in pixels, it is simpler if the post-earthquake affected area is indicated in the same color as the urban area before the earthquake’s occurrence.” (Line 293-297). 

Submission Date

11 December 2018

Date of this review

22 Dec 2018 13:23:39

Round 2

Reviewer 2 Report

The manuscript applies Artificial Intelligence to the analysis of post-earthquake remote sensing imagery in order to map the damage due to the 2018 M7.9 Palu (Indonesia) earthquake. Apparently, the authors have substantially revised the original manuscript according to the last-round review, both in terms of English expression, and scientific contents. I think the it is now close enough to be accepted for publication. Here I only have a few minor suggestions. 

L5: ‘An Mw 7.5 earthquake’: I think USGS registered it as Mw 7.5, and ‘A’ not ‘An’, similar thing at L21.

L48: “Historically, this region has hosted several large earthquakes, the largest event being an Mw 7.9…”  --- “Historically, this region has hosted several large earthquakes, with the largest event being an Mw 7.9”

L56: ‘a single fault method’  what is ‘a single fault method’? a source model consisting only one fault or others?

L59 ‘with higher tsunami run-ups than are straight or cape-shaped shorelines’ --- ‘with higher tsunami run-ups than straight or cape-shaped shorelines’

L65: “for obtaining accurate information”: accurate information of what? I think a specific word here is necessary. 

Author Response

ANSWER SHEET

REVIEWER 2

Open Review

English language and style

( ) Extensive editing of English language and style required 
( ) Moderate English changes required 
(x) English language and style are fine/minor spell check required 
( ) I don't feel qualified to judge about the English language and style 

Yes

Can   be improved

Must   be improved

Not   applicable

Does   the introduction provide sufficient background and include all relevant   references?

(x)

(   )

(   )

(   )

Is   the research design appropriate?

(x)

(   )

(   )

(   )

Are   the methods adequately described?

(   )

(x)

(   )

(   )

Are   the results clearly presented?

(x)

(   )

(   )

(   )

Are   the conclusions supported by the results?

(x)

(   )

(   )

(   )

Comments and Suggestions for Authors

The manuscript applies Artificial Intelligence to the analysis of post-earthquake remote sensing imagery in order to map the damage due to the 2018 M7.9 Palu (Indonesia) earthquake. Apparently, the authors have substantially revised the original manuscript according to the last-round review, both in terms of English expression, and scientific contents. I think it is now close enough to be accepted for publication. Here I only have a few minor suggestions. 

L5: ‘An Mw 7.5 earthquake’: I think USGS registered it as Mw 7.5, and ‘A’ not ‘An’, similar thing at L21.

Answer: Thank you for your comments. The authors edit the sentence from ‘A’ to ‘An’ in line 5 and line 21. USGS registered the magnitude as Mw 7.5 in 28 September 2018 as can be seen in [1], but the authors referred to related Indonesia’s institute, such as Indonesia’s Meteorological, Climatological, and Geophysical Agency (BMKG) [2] , Indonesia’s National Board for Disaster Management [3], Indonesia Institute of Science [4], and ASEAN Coordinating Centre for Humanitarian Assistance on disaster management [5] who release an update towards the earthquakes which can be seen in the links below.

L48: “Historically, this region has hosted several large earthquakes, the largest event being an Mw 7.9…”  --- “Historically, this region has hosted several large earthquakes, with the largest event being an Mw 7.9”

Answer: Thank you for your comments. I edit the sentence as can be seen in Line 48 such as “Historically, this region has hosted several large earthquakes, with the largest event being an Mw 7.9 earthquake in January 1996 at Toli-Toli city [6], around 100 km north of the epicenter of this most recent event”

L56: ‘a single fault method ’what is ‘a single fault method’? a source model consisting only one fault or others?

Answer: Thank you for your comments. In this study we did not use single fault method, so the author did not have in-depth understaning in this method. The author mentioned about the single fault method in refers to Rahmadaningsi et al., (2018) in the introduction part. From the citation reference, we focused about the result of Rahmadaningsi et al., (2018) that expained the bay-shaped shorelines are associated with higher tsunami run-ups than straight or cape-shaped shorelines (line 58-59). Rahmadaningsi et al., (2018) used single fault method to observe the distribution of main shock, foreshock, and aftershock, and it is suitable to see the vertical displacement.

L59 ‘with higher tsunami run-ups than are straight or cape-shaped shorelines’ --- ‘with higher tsunami run-ups than straight or cape-shaped shorelines’

Answer: Thank you for your comments. I edit the sentence as can be seen in Line 59 such as ”The result shows that bay-shaped shorelines are associated with higher tsunami run-ups than straight or cape-shaped shorelines.”

L65: “for obtaining accurate information”: accurate information of what? I think a specific word here is necessary. 

Answer: Thank you for your comments. I add some words to address this issue.

“Two predominant methods are available for obtaining accurate information of post-earthquake..” (Line 65-66)

Links

[1] https://www.usgs.gov/news/magnitude-75-earthquake-near-palu-indonesia

[2] https://www.bmkg.go.id/artikel/?p=pemodelan-distribusi-slip-gempabumi-palu-mw-7-4-tanggal-28-september-2018&tag=&lang=ID

[3] https://sites.google.com/view/gempadonggala/beranda

[4]  http://lipi.go.id/siaranpress/analisis-lipi-untuk-gempa-dan-tsunami-indonesia-/21318

[5] https://ahacentre.org/wp-content/uploads/2018/10/AHA-Situation_Update-no12-Sulawesi-EQ-rev.pdf

Reviewer 3 Report

The comments concerning the rationalism and the references has been taken into account and improved. Moreover the English language has been improved and the sections and the figures have been rearranged.

Author Response

Open Review        

English language and style

                                   ( )                                            Extensive editing of English language and style required                                    
                                   ( )                                            Moderate English changes required                                    
                                   (x)                                            English language and style are fine/minor spell check required                                    
                                   ( )                                            I don't feel qualified to judge about the English language and style                                    

YesCan be improvedMust be improvedNot applicable
Does the introduction provide sufficient background and include all relevant references?(x)( )( )( )
Is the research design appropriate?(x)( )( )( )
Are the methods adequately described?(x)( )( )( )
Are the results clearly presented?(x)( )( )( )
Are the conclusions supported by the results?(x)( )( )( )

Comments and Suggestions for Authors

The comments concerning the rationalism and the references has been  taken into account and improved. Moreover the English language has been  improved and the sections and the figures have been rearranged.

Submission Date

11 December 2018

Date of this review

21 Jan 2019 12:20:18

Answer: Thank you for your good comments for our modified manuscript.